# Inhibition of Indigoidine Synthesis as a High-Throughput Colourimetric Screen for Antibiotics Targeting the Essential *Mycobacterium tuberculosis* Phosphopantetheinyl Transferase PptT

**DOI:** 10.3390/pharmaceutics13071066

**Published:** 2021-07-12

**Authors:** Alistair S. Brown, Jeremy G. Owen, James Jung, Edward N. Baker, David F. Ackerley

**Affiliations:** 1School of Biological Sciences, Victoria University of Wellington, Wellington 6012, New Zealand; alistair.brown@vuw.ac.nz (A.S.B.); jeremy.owen@vuw.ac.nz (J.G.O.); 2Centre for Biodiscovery, Victoria University of Wellington, Wellington 6012, New Zealand; 3Maurice Wilkins Centre for Molecular Biodiscovery, Auckland 1142, New Zealand; jjung@genzentrum.lmu.de (J.J.); en.baker@auckland.ac.nz (E.N.B.); 4School of Biological Sciences, University of Auckland, Auckland 1142, New Zealand

**Keywords:** PPTase, NRPS, indigoidine, PptT, antibiotic screening

## Abstract

A recently-validated and underexplored drug target in *Mycobacterium tuberculosis* is PptT, an essential phosphopantetheinyl transferase (PPTase) that plays a critical role in activating enzymes for both primary and secondary metabolism. PptT possesses a deep binding pocket that does not readily accept labelled coenzyme A analogues that have previously been used to screen for PPTase inhibitors. Here we report on the development of a high throughput, colourimetric screen that monitors the PptT-mediated activation of the non-ribosomal peptide synthetase BpsA to a blue pigment (indigoidine) synthesising form in vitro. This screen uses unadulterated coenzyme A, avoiding analogues that may interfere with inhibitor binding, and requires only a single-endpoint measurement. We benchmark the screen using the well-characterised Library of Pharmaceutically Active Compounds (LOPAC^1280^) collection and show that it is both sensitive and able to distinguish weak from strong inhibitors. We further show that the BpsA assay can be applied to quantify the level of inhibition and generate consistent EC_50_ data. We anticipate these tools will facilitate both the screening of established chemical collections to identify new anti-mycobacterial drug leads and to guide the exploration of structure-activity landscapes to improve existing PPTase inhibitors.

## 1. Introduction

Despite diminishing rates of infection, the disease burden of *Mycobacterium tuberculosis* remains high, with 2019 seeing approximately 10 million people infected and 1.4 million deaths worldwide [1]. The emergence of drug-resistant strains of *M. tuberculosis* coupled with long treatment times has resulted in a pressing need for new therapeutics [2]. *M. tuberculosis* is difficult to treat effectively, in part due to its lipid-rich cell wall and envelope, which contain a diversity of unusual lipids that help it to survive and evade the host immune system [3,4,5]. Mega-synthetases, including the fatty acid synthetase (FAS) I and II systems and polyketide synthetases (PKSs), play crucial roles in the biosynthesis of these lipids [6]. A further mega-synthetase family, the non-ribosomal peptide synthetases (NRPSs), is required to produce the important virulence factor mycobactin [7]. Each of these mega-synthetases requires the attachment of a phosphopantetheinyl (Ppant) arm to one or more carrier protein (CP) domain(s) to convert them from an inactive *apo* to an active *holo* form, a post-translational modification that is essential for functionality [8].

The attachment of the Ppant arm is catalysed by an enzyme superfamily called the 4′-phosphopantetheinyl transferases (PPTases), which in prokaryotes fall into two broad classes that differ in their structure and substrate specificity [8]. Type I (or AcpS type) PPTases are homotrimers that have a narrow substrate specificity and typically recognise acyl carrier protein (ACP) domains present in the FAS-I and FAS-II systems. Type II (or Sfp type) PPTases tend to be pseudodimers, have a much broader substrate specificity and typically activate ACP, peptidyl carrier protein (PCP) and aryl carrier protein (ArCP) domains present in PKSs and NRPSs [8]. Due to their lynchpin roles in both primary and secondary metabolism, many PPTases are essential [8] and have been identified as promising drug targets [9].

*M. tuberculosis* possesses both a Type I PPTase (AcpS) and a Type II PPTase (PptT) [10]. Although it activates the FAS-1 system [11], the essential nature of AcpS has not been confirmed in *M. tuberculosis* [10,12]. Conversely, PptT, which governs the activation of at least 18 PKSs [13], three NRPSs involved in the biosynthesis of the siderophore mycobactin [14] and AcpM (the standalone CP in the FAS-II system [11]), has been confirmed as essential for *M. tuberculosis* growth in vitro [12,13] and in murine models [13]. Importantly for drug targeting, even partial inhibition of PptT can be enough to kill *M. tuberculosis* [13]. This is likely because a Ppant hydrolase (PptH) that removes the Ppant from carrier proteins is expressed in the same operon as PptT, thereby restricting the ability of *M. tuberculosis* to upregulate PptT without also increasing PptH to detrimental levels [15]. PptT is a pseudodimer and has a broadly similar α/β fold to other crystallised Type-II PPTases with some minor variations, one of the most significant being that the Ppant arm extends into a deep hydrophobic pocket in the binding pocket [16,17]. By way of contrast, in the crystal structure of the well-characterised Type II PPTase, Sfp from *Bacillus subtilis*, the Ppant arm directs into the solvent [18] (Figure 1A).

Several high throughput screens to identify bacterial PPTase inhibitors have been developed that monitor the binding of fluorescently labelled coenzyme A (CoA) to a carrier protein or peptide, using either fluorescent polarisation or FRET-based detection [9,19,20,21]. However, rather than directly targeting the PPTase of a pathogenic species, these screens have all employed the canonical PPTase Sfp from *B. subtilis* as a surrogate. This is problematic for discovering inhibitors of *M. tuberculosis* PptT, as it does not accept fluorescent CoA analogues as readily as Sfp [13], due to its deeper binding pocket (Figure 1A) [16,17]. It is also noteworthy that 8918, a promising PptT inhibitor that was recently identified in a whole-organism screen against *M. tuberculosis*, binds to the PptT active site, close to the Ppant arm [15]. Fluorescent analogues of CoA, which are substantially bulkier than native CoA, may therefore exclude otherwise promising inhibitors from the active site. With this in mind, we considered it important to develop a cost-effective direct screen against PptT that uses unadulterated CoA as a substrate.

We have previously shown that the NRPS BpsA (blue pigment synthetase A) could be used to assess the relative levels of inhibition of Sfp and two *Pseudomonas* Type II bacterial PPTases by the generic inhibitor 6-nitroso-1,2-benzopyrone [22]. BpsA is a single-module NRPS that in vitro can convert two molecules of L-glutamine into the blue pigment indigoidine, provided it can been activated to the *holo* form by a co-incubated PPTase (Figure 1B) [23]. Here we demonstrate that recombinant BpsA purified in the *apo* form can be used to provide a robust and high-throughput screen for compounds that inhibit PptT from activating BpsA.

## 2. Materials and Methods

### 2.1. Materials and Reagents

Unless otherwise stated, chemicals, media and reagents used in this study were supplied by Sigma-Aldrich (St Louis, MO, USA), Thermo Fisher Scientific (Waltham, MA, USA), Duchefa Biochemie (BH Haarlem, Netherlands) or New England Biolabs (Ipswich, MA, USA). Sanguinarine chloride for kinetic screening was supplied by Sapphire Biosciences (Redfern, NSW, Australia).

### 2.2. Plasmid Construction

Construction of the BpsA expression plasmid pCDFDUET1::*bpsA* was described previously [22]. Construction of NOHISPET::*pcpS*, which expresses an untagged version of the broad-spectrum PPTase PcpS to covert *apo*-BpsA to *holo*-BpsA and not co-purify during Ni-NTA chromatography, was as previously described [25]. pET28a(+)::*pptT* was constructed by amplifying *pptT* from *M. tuberculosis* H37Ra genomic DNA using the primers CCCCCATATGGACGGTAGGCACGCTG and CCCCCTCGAGTCATAGCACGATCGCGGT (restriction sites underlined). The amplified gene was ligated into pET28a(+) between the *Nde*I and *Xho*I restriction sites.

### 2.3. Protein Expression

All protein expression used *E. coli* BL21(DE3) Δ*entD* [26] as an expression strain. Cultures were grown in lysogeny broth (LB) with plasmid appropriate antibiotics (spectinomycin 100 µg/mL, kanamycin 50 µg/mL). Fresh 400 mL expression cultures were inoculated to an A_600_ of 0.1 overnight and incubated at 37 °C, 200 rpm until an A_600_ of between 0.6 and 0.8 was obtained. The cultures were then chilled on ice for approximately 30 min. Protein expression was induced with the addition of IPTG to a final concentration of 0.5 mM. The cultures were then incubated for 24 h at 18 °C, 200 rpm before harvesting by centrifugation (2700× *g*, 20 min, 4 °C). The cell pellets were then stored at −80 °C until needed.

### 2.4. Protein Purification

PPTases were purified using standard Tris-Cl Ni-NTA chromatography buffers with the following modification: the bind, wash and elute buffers were supplemented with 25% *v*/*v* glycerol to enhance protein stability. The eluted PPTases were then desalted using a HiTrap desalting column (Cytiva, Marlborough, MA, USA) with a desalting buffer of 50 mM Tris-Cl (pH 7.8) and 12.5% glycerol (*v*/*v*). The buffer composition was then adjusted to a final concentration of 40% (*v*/*v*) glycerol, and aliquots were stored at −80 °C until needed. *Apo* and *holo*-BpsA were purified as previously described [25].

### 2.5. Optimisation of Enzyme Concentrations

To determine the optimal concentration of BpsA for the detection of PptT inhibition, 30 µL of reaction mix comprising 50 mM Tris-Cl (pH 8.0), 10 mM MgCl_2_ and 5 mM ATP was added to individual wells of a standard 96 well plate. Next, 10 µL *holo*-BpsA (purified and activated as previously described [25]) was added to each well to give a range of final concentrations (0–1 µM). To initiate indigoidine synthesis, 10 µL of 5 mM L-glutamine was added to each well. The plate was then shaken at 1000 rpm for 10 s and incubated at room temperature for 1 h. The indigoidine was then resolubilised by the addition of 200 µL DMSO and then incubated at 37 °C with shaking at 200 rpm for 0.5 h. A Perkin Elmer Enspire Plate reader was then used to record the absorbance readings at 590 nm.

To determine the optimal concentration of PptT for the detection of inhibitors, 10 µL of PPTase mix comprising 50 mM Tris-Cl (pH 8.0) and PptT at a range of concentrations (0–0.5 µM) was added to individual wells of a standard 96 well plate. To initiate indigoidine synthesis, 40 µL of a reaction mix comprising 50 mM Tris-Cl (pH 8.0), 10 mM MgCl_2_, 5 mM ATP, 5 mM L-glutamine, 10 µM CoA and 0.6 µM *apo*-BpsA was added to each well. The plate was then shaken at 1000 rpm for 10 s and incubated at room temperature for 1 h. The indigoidine was then resolubilised by the addition of 200 µL DMSO and incubated at 37 °C with shaking at 200 rpm for 0.5 h. A Perkin Elmer Enspire plate reader was then used to record the absorbance of each well at 590 nm.

### 2.6. Z-Factor Calculation

To determine the *Z-factor* of the screen, 80 reactions were established in columns 2–11 of a standard 96 well-plate. Each was prepared from 30 µL of reaction mix comprising 50 mM Tris-Cl (pH 8.0), 0.01% (*v*/*v*) Triton-X, 10 mM MgCl_2_, 5 mM ATP, 5 mM L-glutamine and 10 µM CoA. Next, 10 µL PPTase mix containing either 0.4 µM PptT and 50 mM Tris-Cl (pH 8.0) (or else unadulterated 50 mM Tris-Cl (pH 8.0) as a negative control) was added to each well. To initiate indigoidine synthesis, 10 µL of 0.6 µM *apo*-BpsA in 50 mM Tris-Cl (pH 8.0) was added to each well. The plate was then shaken at 1000 rpm for 10 s and incubated at room temperature for 1 h. The indigoidine was then resolubilised by the addition of 200 µL DMSO and incubated at 37 °C with shaking at 200 rpm for 0.5 h. A Perkin Elmer Enspire plate reader was then used to record the absorbance readings at 590 nm. To calculate the *Z-factor*, the following equation was used (derived from the method of Zhang et al. [27]), with σ representing the standard deviation, µ representing the means and *ρ* representing the positive and n the negative control.
$$Z\;factor=1-\frac{3\left(\sigma_{\rho }+\sigma_{n}\right)}{\left|\mu_{\rho }-\mu_{n}\right|}$$

### 2.7. Screening of the LOPAC^1280^ Collection and EC_50_ Calculations

The Sigma-Aldrich LOPAC^1280^ collection was first diluted to a working stock of 500 mM per compound. To screen the collection, 88 reactions at a time were established in columns 1–11 of a standard 96 well-plate. Each was prepared from 30 µL of reaction mix comprising 50 mM Tris-Cl (pH 8.0), 0.01% (*v*/*v*) Triton-X, 10 mM MgCl_2_, 5 mM ATP, 5 mM L-glutamine and 10 µM CoA. Next, a CyBio liquid handler was used to pin 2 µL of compound into columns 2–11 and 2 µL of DMSO into column 1. To screen for PptT inhibitors, 10 µL of PPTase mix comprising a freshly-prepared stock of 0.4 µM PptT and 50 mM Tris-Cl (pH 8.0) was added to wells A1–C1 and columns 2–11 (see Appendix A). To initiate indigoidine synthesis, 10 µL of 0.6 µM *apo*-BpsA in 50 mM Tris-Cl (pH 8.0) was added to each well. To screen for *holo*-BpsA inhibitors, an equivalent screening format was established only using 0.6 µM *holo*-BpsA and with no PPTase mix added. Each plate was then shaken at 1000 rpm for 10 s and incubated at room temperature for 1 h. Indigoidine was then resolubilised by the addition of 200 µL DMSO and incubation at 37 °C with shaking at 200 rpm for 0.5 h. A Perkin Elmer Enspire plate reader was then used to record the absorbance readings at 590 nm. The Python package Pandas [28] was used to process the data, which was then visualised using the Python package Seaborn [29].

To calculate the EC_50_ values of inhibitors, a two-fold serial dilution between 40 µM and 0.625 µM of each compound was established in 30 µL reaction mixes comprising 50 mM Tris-Cl (pH 8.0), triton-X 0.01% (*v*/*v*), 10 mM MgCl_2_, 5 mM ATP, 5 mM L-glutamine and 10 µM CoA in replicate rows of a 96 well plate. To screen for PptT inhibitors, 10 µL of PPTase mix comprising a freshly-thawed stock of 0.4 µM PptT in 50 mM Tris-Cl (pH 8.0) was added. To initiate indigoidine synthesis, 10 µL of 0.6 µM *apo*-BpsA in 50 mM Tris-Cl (pH 8.0) was added to each well. The plate was then shaken at 1000 rpm for 10 s and incubated at room temperature for 1 h. Indigoidine was resolubilised by the addition of 200 µL DMSO and incubation at 37 °C with shaking at 200 rpm for 0.5 h. A Perkin Elmer Enspire plate reader was then used to record the absorbance readings at 590 nm. The percentage inhibition at each compound concentration was then compared to a DMSO only control, and GraphPad Prism 9.0 was used to fit a four parameter dose–response curve.

### 2.8. Kinetic Analysis of PptT Inhibition

To calculate the inhibition of PptT by 6-nitroso-1,2-benzopyrone and sanguinarine chloride, a serial dilution between 200 µM and 0.2 µM was established for each compound in 30 µL 50 mM Tris-Cl (pH 8.0) and Triton-X 0.01% (*v*/*v*) across individual rows of a standard 96 well plate. Next, 35 µL of a PPTase mix comprising a freshly-thawed stock of 0.4 µM PptT, 50 mM Tris-Cl (pH 8.0) and Triton-X 0.01% (*v*/*v*) was added to each well. To initiate indigoidine synthesis, a 35 µL reaction mix comprising 0.01% (*v*/*v*) Triton-X, 50 mM Tris-Cl (pH 8.0), 10 mM MgCl_2_, 5 mM ATP, 5 mM L-glutamine, 10 µM CoA and 0.6 µM apo-BpsA was added to each well. The plate was then shaken at 1000 rpm for 10 s, and absorbance readings at 590 nm were read every 20 s. To determine the EC_50_ values, the maximum indigoidine synthesis velocities were determined as described previously [22], and four-parameter dose–response curves were fitted using GraphPad Prism version 9.0.

## 3. Results

To determine whether BpsA could be used to screen for inhibitors of PptT, it was first necessary to show that PptT is capable of activating BpsA from the inactive *apo* to active *holo* form. BpsA was purified in the *apo* form from the Type II PPTase null *E. coli* strain BL21 Δ*entD* [26] and co-incubated with purified PptT. We and others [16,17,30] have found PptT to be quite unstable in aqueous solutions, with it rapidly aggregating and losing activity. This necessitated freshly thawed PptT (purified and stored on glycerol at −80 °C) to be used for each assay. Nonetheless, PptT was capable of rapidly activating *apo*-BpsA in vitro, yielding a typical indigoidine synthesis curve that reflects an initial burst of indigoidine synthesis followed by subsequent indigoidine precipitation [22] (Figure 1C).

In our previous demonstration that BpsA could be used to quantify the strength of PPTase inhibition by the broad-specificity PPTase inhibitor 6-nitroso-1,2-benzopyrone, we monitored the change in velocity of indigoidine synthesis via continuous kinetic measurements [22]. We have since developed BpsA as a biosensor to measure glutamine [25] and ATP [31] and in so doing observed that end-point measurements post-solubilisation of the total indigoidine in 80% DMSO yielded more consistent data. To implement a similar approach here to quantify inhibition of PptT, we first assayed pre-activated *holo*-BpsA at a concentration range from 0 to 1 µM in 50 µL reactions. This enabled us to identify 0.6 µM BpsA as an enzyme concentration that could provide a robust signal within the linear range of the assay, in a volume that could accommodate the addition of DMSO to 80% (*v*/*v*) without exceeding the well capacity of a standard 96-well plate (Figure 2A). We then identified 0.4 µM PptT as a suitable concentration of target PPTase that, in the absence of inhibitor, induced a strong indigoidine synthesis signal when co-incubated with 0.6 µM *apo*-BpsA (Figure 2B).

To validate the performance of the assay under conditions similar to a high throughput screen, we compared 80 reactions in a 96 well plate. A screening master mix comprising buffer, all requisite substrates (CoA, ATP and L-glutamine) and 0.01% Triton X-100 (to minimise nonspecific inhibitory activity from compounds that induce protein aggregation [32]) was aliquoted into each well. Next, PptT was added to 40 of the wells (to a final concentration of 0.4 µM), while the other 40 wells received only buffer to simulate reactions in which PptT had been completely inhibited. Finally, *apo*-BpsA was added to each well (to a final concentration of 0.6 µM) to initiate indigoidine synthesis, where possible. Following incubation for 1 h at room temperature, each 50 µL reaction was stopped by the addition of 200 µL of DMSO to solubilise the indigoidine, after which the A_590_ was measured (Figure 2C). The calculated Z’-factor for the assay was 0.77. This is a key measure of quality for high-throughput screens that derives from the separation of positive and negative control data; a Z’-factor above 0.5 represents 12 standard deviations of separation between the controls and hence a highly robust screen [27].

Having established the assay parameters, we then screened 1280 drug-like compounds from the Sigma-Aldrich library of pharmacologically active compounds (LOPAC^1280^) to identify inhibitors of PptT (a schematic of the complete high-throughput screening procedure is presented in Appendix A). The LOPAC^1280^ collection is widely used for screen validation studies and has previously been found to contain a diverse range of inhibitors of *B. subtilis* Sfp [19], giving us confidence that it was likely to contain a diversity of PptT inhibitors also. Each screening plate contained 80 compounds arrayed in columns 2–11. Three no-inhibitor positive controls and three negative controls lacking PptT were arrayed in column 1 of the screening plate, while column 12 was left empty (Appendix A). The percentage activity for each well was then calculated relative to the average of the three positive controls located on each plate, revealing that 21 compounds caused ≥50% reduction in indigoidine levels (Figure 3B). Position effects were evident, owing to the instability of PptT upon addition to the aqueous reaction mix (in a row-by-row fashion using a multi-well pipette), which yielded a wave-like pattern of A_590_ readings from row to row (Appendix A). This phenomenon was not observed in a counter-screen that was conducted using pre-activated *holo-*BpsA, to eliminate any compounds that were inhibiting BpsA rather than PptT (Appendix A). While it is possible that variation in the levels of soluble PptT may have confounded identification of some weak PptT inhibitors, most of the 21 “hits” yielded data that were clearly separated from the baseline (Figure 3B, dashed line).

In counter-screening the LOPAC^1280^ collection against pre-activated *holo*-BpsA (Appendix A), four compounds were identified as inhibiting indigoidine synthesis rather than being PptT-specific: ebselen, suramin hexasodium, chlorprothixene hydrochloride and 4-chloromercuribenzoic acid. Chlorprothixene hydrochloride did not register as an inhibitor of PptT and may have been a false positive in this counter-screen, while the other three were discarded from further consideration as likely pan assay interference compounds (PAINS) [33].

We next quantified the inhibition conferred by each of the 18 remaining compounds, using two-fold serial dilutions (40 µM to 0.3 µM) of each compound added to a PptT/*apo*-BpsA reaction mix to generate EC_50_ values (Appendix A). In one case (8-hydroxy-DPAT hydrobromide), no inhibitory activity was observed, indicating this was likely a false positive from our initial screen, while for nine of the remaining cases, the level of inhibition was insufficient to calculate meaningful EC_50_ values. Ultimately, we were able to confirm and quantify eight effective inhibitors of PptT (Table 1). Other than tyrphostin AG 537 and tyrphostin AG 538, all of these compounds had previously been reported as inhibitors of the canonical PPTase Sfp from *B. subtilis* [32].

In other work, we recently screened the LOPAC^1280^ collection against the SARS-CoV-2 protease using a FRET-based reporter and noticed that 8 of our 17 candidate PptT inhibitors (PD 404, 182, Disulfiram, U-73122, 6-nitroso-1,2-benzopyrone, aurintricarboxylic acid, sanguinarine chloride, Bay 11-7085 and 6-hydroxyl-DL-DOPA) also gave hits as strong protease inhibitors [34]. This suggested to us that many of these compounds may be generic enzyme inhibitors rather than specific for PptT. To further probe the specificity of hit compounds for PptT, we re-screened each for inhibition of *holo*-BpsA at a higher concentration of 40 µM (Appendix A). In this higher-concentration counter-screen, SCH 202626, previously identified as a broad-spectrum PPTase inhibitor [19], became noticeably inhibitory of *holo*-BpsA. Only five compounds exhibited less than 15% inhibition of *holo-*BpsA: 6-nitroso-1,2-benzopyrone, 6-hydroxy-DL-DOPA, 7,7-dimethyl-(5Z,8Z)-eicosadienoic acid, disulfiram and sanguinarine chloride.

We next sought to benchmark our DMSO-resolubilisation end-point assay against the kinetic assay we had previously described [22]. For this, we selected two compounds with different inhibition profiles, namely 6-nitroso-1,2-benzopyrone, which has been shown to have activity in vitro against numerous PPTases [19,22,35] and in vivo against the fungus *Aspergillus fumigatus* in a PPTase dependant manner [36], and the weaker inhibitor sanguinarine chloride, which has been previously reported as an inhibitor of both Sfp from *B. subtilis* [21] and *M. tuberculosis* PptT [16]. We used identical enzymatic concentrations across both assays to permit a direct comparison. In the kinetic assay, we derived an EC_50_ value of 4.4 ± 1.2 µM for 6-nitroso-1,2-benzopyrone (Appendix A), which is similar to the EC_50_ value of 8.3 µM we calculated from our endpoint assay. However, the kinetic assay data for sanguinarine chloride was more variable (Appendix A), and GraphPad Prism 9.0 was unable to fit a robust curve. Based on this, we conclude that the endpoint assay is more sensitive and reproducible than the kinetic assay, at least at the tested enzyme concentrations.

## 4. Discussion

Here, we report on the development, optimisation and validation of a simple high-throughput assay to rapidly identify inhibitors of the promising antibiotic target PptT from *M. tuberculosis*. We validated the assay by screening the LOPAC^1280^ collection from Sigma-Aldrich. To the best of our knowledge, this is the first reported direct chemical library screen for inhibitors of PptT. From this screen we identified 18 inhibitors, but further investigation indicated that ten of these were non-specific compounds that also inhibited *holo-*BpsA when added at higher concentrations. Of the remaining eight, two were novel PPTase inhibitors that may be specific for PptT, while six had previously been identified as inhibitors of the *B. subtilis* canonical Type II PPTase Sfp, providing evidence by association for the efficacy of our screen. Finally, we compared our optimised end-point assay against a kinetic BpsA assay we had employed earlier [22] and concluded that the optimised assay is more amenable to high-throughput screening.

Two other assays have been developed to evaluate possible inhibitors of PptT. The first uses a scintillation proximity assay that is substantially more complex than our simple colourimetric BpsA assay, requiring radiolabelled CoA and a scintillation counting machine, and validation against PPTase inhibitors has not yet been reported [13]. The second assay uses rhodamine-labelled CoA to provide a more accessible fluorometric output [16]. This assay has been tested against a small panel of PptT inhibitors, but it is unclear if it would be able to detect inhibitors such as 8918. While 8918 is the best PptT inhibitor reported to date [15], it binds immediately alongside CoA in the active site and might be precluded from binding by a conjugated rhodamine at that position. As our screen uses unadulterated CoA, this would not be a concern.

PPTases have been discussed in the literature as promising drug targets for almost two decades [37]. Despite this, limited progress has been made on progressing inhibitors through drug development pipelines. Thus far the two most promising PPTase inhibitor drug candidates are ML-267, which was identified after high-throughput screening of approximately 330,000 compounds against *B. subtilis* Sfp [38], but has not yet had efficacy reported against other PPTases; and 8918, which was identified from a whole-cell screen of *M. tuberculosis* HR7Rv against approximately 90,000 small molecules, with PptT subsequently identified as the drug target [15]. While whole cell screens offer the major advantage that biological activity is guaranteed, at least under the assay conditions employed, target identification is often far more difficult, and high-proportions of time-wasting promiscuous “nuisance compounds” are frequently recovered [39]. Moreover, for mycobacterial screens it has been reported that variations in media can substantially alter the spectrum of inhibitors identified from a given chemical collection, raising concerns as to how well host-relevant physicochemical conditions are represented [40]. Conversely, target-based screens for *M. tuberculosis* offer substantial promise in enabling selection of a suitable target upfront [41] and facilitating rational exploration of structure–activity relationships [42]. Our assay therefore meets a need for a simple high-throughput assay that can be applied to directly screen for inhibitors of PptT or other Type II PPTases, or to guide the development of next-generation drug leads based on promising scaffolds such as ML-267 and 8918. Recently, we have also shown that the PCP-domain of BpsA can be rapidly evolved using error-prone mutagenesis to be recognised by other Type II PPTases that initially cannot activate BpsA [35]. This provides a platform to enable the rapid integration of multiple PPTases for inhibitory activity, or to counter-screen the endogenous human PPTase to eliminate drug candidates likely to have undesirable off-target effects. Collectively, we hope this toolbox will prove useful in guiding future medicinal chemistry efforts to design or identify broad spectrum PPTase inhibitors.

## Figures and Tables

**Figure 1 pharmaceutics-13-01066-f001:**
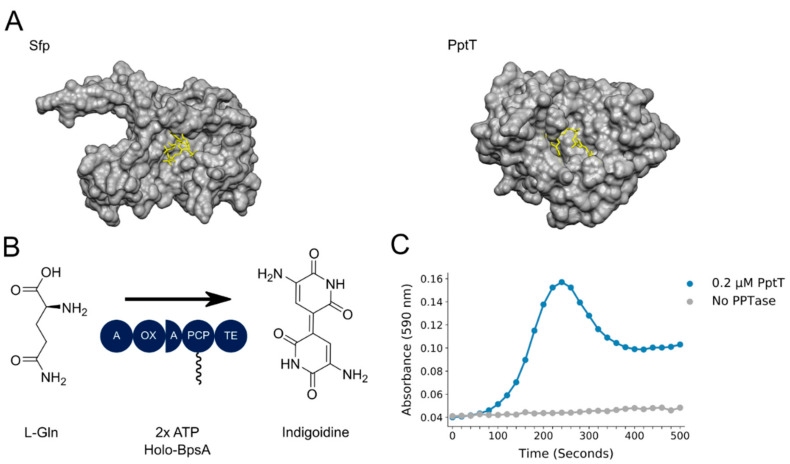
(**A**) Crystal structures of Sfp and PptT, highlighting the orientation of CoA: CoA (yellow) is orientated towards the solvent in the crystal structure of Sfp (PDB: 1QR0), while in our structure of PptT (PDB:4QVH), it is encapsulated by the binding pocket. Visualisation was performed using Chimera [24]. (**B**) Schematic showing biosynthesis of indigoidine by *holo*-BpsA: *Holo*-BpsA synthesises indigoidine from two molecules of L-glutamine in an ATP powered reaction. When indigoidine is solubilised in DMSO, it has a characteristic deep blue colouring that can readily be detected by monitoring absorbance at 590 nm. (**C**) Characteristic curve of indigoidine biosynthesis during the activation of *apo*-BpsA by PptT: Indigoidine production in an aqueous solution yields a sigmoidal curve of absorbance at 590 nm until it reaches saturation and precipitates, after which the A_590_ drops.

**Figure 2 pharmaceutics-13-01066-f002:**
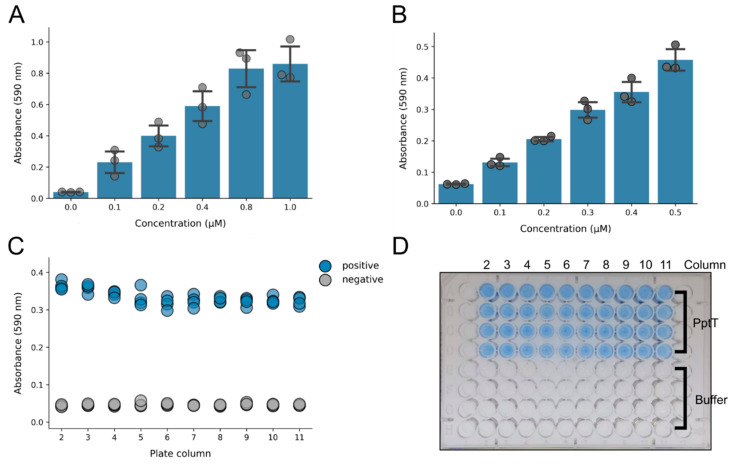
Optimisation of assay parameters: (**A**) increasing concentrations of *holo*-BpsA were added to a reaction mix containing 10 mM MgCl_2_, 5 mM ATP and 50 mM Tris-Cl (pH 8.0). The reaction was then initiated by addition of 5 mM L-glutamine and incubated for 1 h. The indigoidine was then resolubilised by the addition of DMSO, which gave a linear increase in absorbance until the assay began to saturate at an A_590_ of approximately 1.0. Data are the averages of three independent replicates and error bars represent one standard deviation. (**B**) Increasing concentrations of PptT were added to a reaction mix containing 10 mM MgCl_2,_ 5 mM ATP, 5 mM L-glutamine, 0.6 µM apo-BpsA and 10 µM CoA. The reaction was incubated for 1 h and then resolubilised by the addition of DMSO, and A_590_ values were recorded. A higher concentration of PptT resulted in an increased absorbance reading, indicating that a single endpoint assay can be used to detect inhibition. Data are the averages of three independent replicates, and error bars represent one standard deviation. (**C**) To determine the Z’ factor of the screen, a reaction mix was established in columns 2–11 comprising 0.01% triton X, 50 mM Tris-Cl (pH 8.0), 10 mM MgCl_2_, 5 mM ATP, 5 mM L-glutamine and 10 µM CoA. Next, PptT (to a final concentration of 0.4 µM) and buffer were added to rows 1–4, while only buffer was added to rows 5–8. To initiate each reaction, 0.6 µM *apo*-BpsA was added. The A_590_ values for rows containing PptT (blue, positive) and the buffer controls (grey, negative) were recorded following a 1 h incubation and subsequent solubilisation with DMSO. (**D**) A photo of the plate from the experiment described for panel C following solubilisation by DMSO. The plate was placed on a light box prior to taking the photo.

**Figure 3 pharmaceutics-13-01066-f003:**
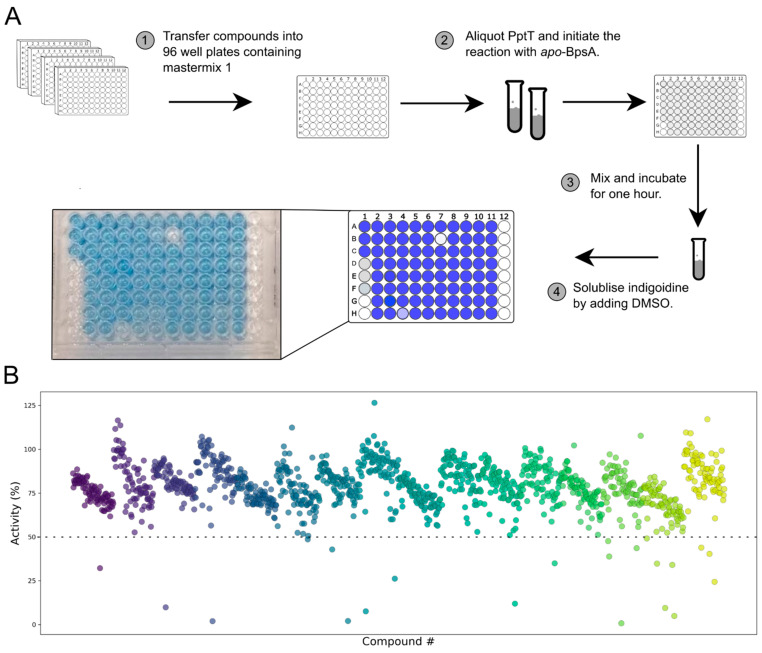
High-throughput screening of the LOPAC^1280^ library: (**A**) schematic diagram illustrating the screening process; (**B**) percentage activity of each compound in the library, with data derived from distinct 96 well plates presented in different colours.

**Table 1 pharmaceutics-13-01066-t001:** PptT inhibitors recovered from BpsA screen. Names and descriptions were derived from the LOPAC^1280^ manifest. EC_50_ averages were calculated from three independent replicates ± standard error.

Name	Known Drug Activities	% Activity in PptT Screen	EC_50_ (µM)
Aurintricarboxylic acid	DNA topoisomerase II inhibitor	32	4.8 ± 0.9
SCH-202676 hydrobromide	Allosteric agonist and antagonist of GPCRs	0.7	5.9 ± 0.1
Tyrphostin AG 537	EGFR protein tyrosine kinase inhibitor	35	5.9 ± 0.5
Disulfiram	Alcohol dehydrogenase inhibitor	9	6.0 ± 1.5
Tyrphostin AG 538	(IGF-1) receptor protein tyrosine kinase inhibitor	24	6.4 ± 0.7
Bay 11-7085	Inhibitor of NFκB	10	7.4 ± 0.4
U-73122	Phospholipase C and A2 inhibitor	44	8.1 ± 0.5
6-Nitroso-1,2-benzopyrone	Poly(ADP-ribose) ligand	12	8.3 ± 1.7
6-Hydroxy-DL-DOPA	Precursor of the catecholaminergic neurotoxin	26	N.D. ^1^
Tyrphostin AG 808	Protein tyrosine kinase inhibitor	34	N.D. ^1^
I-OMe-Tyrphostin AG 538	(IGF-1) receptor protein tyrosine kinase inhibitor	40	N.D. ^1^
Reactive Blue 2	P2Y receptor antagonist	38	N.D. ^1^
GW5074	cRaf1 kinase inhibitor	43	N.D. ^1^
PD 404, 182	KDO-8-P synthase inhibitor	35	N.D. ^1^
Sanguinarine chloride	Inhibitor of ATPase	44	N.D. ^1^
7,7-Dimethyl-(5z,8z)-eicosadienoic acid	Phospholipase A2 and lipoxygenase inhibitor	49	N.D. ^1^
Rottlerin	PKC and CaM kinase III inhibitor	48	N.D. ^1^

^1^ N.D. = not determined, i.e., binding was insufficiently strong to permit a robust curve to be fitted to determine an EC_50_ value.

## Data Availability

The data presented in this study are available on request from the corresponding author.

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
