# Peer review of "Inhibition of Indigoidine Synthesis as a High-Throughput Colourimetric Screen for Antibiotics Targeting the Essential Mycobacterium tuberculosis Phosphopantetheinyl Transferase PptT"

_pharmaceutics, 2021, doi:10.3390/pharmaceutics13071066_

Round 1

Reviewer 1 Report

Since Mycobacterium tuberculosis continues to be among the most dangerous infective organisms, novel antibiotics are still in great demand. In their paper Brown et al. describe a novel screening method based on the specific requirement of this organism for phosphopantetheine in four mega-synthetases. In their assay they use conversion of glutamine to blue-colored indigoidine catalized by the blue pigment synthetase BpsA which requires activation by a 4’-phosphopanthetheinyl transferase (PPTase). Inhibition of the latter by new drugs may deplete Mycobacteria from essential metabolites. Among thousands of compounds they tested from a drug library they identified eight compounds of which two were novel inhibitors of indigoidin synthesis. The assay meets the purpose of successful screening for transferase inhibitors. The method was well developed taking into account possible experimental pitfalls. However, the paper lacks testing of the antibiotic activity of the novel compounds with Mycobacteria. Mycobacteria are notoriously resistant against antibiotics. For this reason a careful inspection of possibly active drugs should be performed before such a selection procedure is started. The simple and straightforward procedure will only be employed by readers if in vivo acivity of the identified transferase inhibitors can be demonstrated. Even then, additional hurdels will show up such as toxicity, poor pharmacokinetics etc. 

Author Response

Please see the attachment (this is the same as our cover letter to the academic editor - we apologise if this is a duplication, but were uncertain whether the reviewer is able to access the cover letter).

Reviewer 2 Report

Authors report on the development, optimisation and validation of a simple high-throughput assay to rapidly identify  inhibitors of the promising antibiotic target PptT from M. tuberculosis.
I believe that this manuscript is appropriate for publication by the Pharmaceuticals without any changes.

Author Response

We thank this reviewer for their positive assessment of our work and recommendation that it be published without changes.

Round 2

Reviewer 1 Report

The concept is now better described and sounds convincing. The authors also give good resaons why they prefer not to evaluate the inhibition of mycrobial cells. A screnung method as simple as descibed in this paper is of great value.